# Reciprocal Changes in miRNA Expression with Pigmentation and Decreased Proliferation Induced in Mouse B16F1 Melanoma Cells by l-Tyrosine and 5-Bromo-2′-Deoxyuridine

**DOI:** 10.3390/ijms22041591

**Published:** 2021-02-05

**Authors:** Hernán Mauricio Rivera, Esther Natalia Muñoz, Daniel Osuna, Mauro Florez, Michael Carvajal, Luis Alberto Gómez

**Affiliations:** 1Department of Medicine, Universidad Nacional de Colombia, Bogotá 111321, Colombia; hmriverae@unal.edu.co (H.M.R.); enmunozro@unal.edu.co (E.N.M.); 2Molecular Physiology Group, Sub-Direction of Scientific and Technological Research, Direction of Public Health Research, National Institute of Health, Bogotá 111321, Colombia; 3Science Department, Universidad Nacional de Colombia, Bogotá 111321, Colombia; daosunag@unal.edu.co (D.O.); maeflorezro@unal.edu.co (M.F.); mscarvajalp@unal.edu.co (M.C.); 4Department of Physiological Sciences, Faculty of Medicine, Universidad Nacional de Colombia, Bogotá 111321, Colombia

**Keywords:** melanoma, miRNAs, l-tyrosine, 5-bromo-2′-deoxyuridine, melanin, senescence, pigmentation

## Abstract

*Background*: Many microRNAs have been identified as critical mediators in the progression of melanoma through its regulation of genes involved in different cellular processes such as melanogenesis, cell cycle control, and senescence. However, microRNAs’ concurrent participation in syngeneic mouse B16F1 melanoma cells simultaneously induced decreased proliferation and differential pigmentation by exposure to 5-Brd-2′-dU (5’Bromo-2-deoxyuridine) and L-Tyr (L-Tyrosine) respectively, is poorly understood. *Aim*: To evaluate changes in the expression of microRNAs and identify which miRNAs in-network may contribute to the functional bases of phenotypes of differential pigmentation and reduction of proliferation in B16F1 melanoma cells exposed to 5-Brd-2′-dU and L-Tyr. *Methods*: Small RNAseq evaluation of the expression profiles of miRNAs in B16F1 melanoma cells exposed to 5-Brd-2′-dU (2.5 μg/mL) and L-Tyr (5 mM), as well as the expression by qRT-PCR of some molecular targets related to melanogenesis, cell cycle, and senescence. By bioinformatic analysis, we constructed network models of regulation and co-expression of microRNAs. *Results*: We confirmed that stimulation or repression of melanogenesis with L-Tyr or 5-Brd-2′-dU, respectively, generated changes in melanin concentration, reduction in proliferation, and changes in expression of microRNAs 470-3p, 470-5p, 30d-5p, 129-5p, 148b-3p, 27b-3p, and 211-5p, which presented patterns of coordinated and reciprocal co-expression, related to changes in melanogenesis through their putative targets *Mitf*, *Tyr* and *Tyrp1*, and control of cell cycle and senescence: *Cyclin D1*, *Cdk2*, *Cdk4*, *p21*, and *p27*. *Conclusions*: These findings provide insights into the molecular biology of melanoma of the way miRNAs are coordinated and reciprocal expression that may operate in a network as molecular bases for understanding changes in pigmentation and decreased proliferation induced in B16F1 melanoma cells exposed to L-Tyr and 5-Brd-2′-dU.

## 1. Introduction

Many transcripts show hundreds of genes change in expression in melanoma progression. These changes involve genetic and epigenetic mechanisms, such as mutations, gene silencing of tumor suppressors through DNA methylation in promoter regions, oncogene activation, chromatin remodeling, and regulation mediated by non-coding RNAs, such as microRNAs (miRNAs) through their regulation of genes involved in different cellular processes such as proliferation, death, cycle control, senescence, angiogenesis, differentiation, and even metastasis [1,2,3,4,5,6,7,8]. miRNA can have hundreds of mRNAs as molecular targets. One mRNA can target several miRNAs, and the researchers will identify variations in the expression of thousands of genes that may result from changes in some miRNAs’ expression patterns [9,10,11,12,13].

Previous works on the mouse melanoma B16F1 cell line showed that 72-h exposure to 5 mM L-Tyr resulted in increased cell size and pigmentation, while exposure to 5-Brd-2′-dU 2.5 µg/mL showed cells with a senescence-associated phenotype and hypopigmentation; however, a decrease in cell growth was reported [14,15]. These changes coincide with the differential expression of hundreds of genes evaluated in this experimental model, including cyclins, cyclin-dependent kinases (CDKs), and CDKs inhibitors [16,17]. In 2008, it was possible to identify two miRNAs (miR-138-5p and miR-470-3p) that presented differential expression, which had *cyclin D1* as a molecular target [18]. However, the differential expression profile of miRNAs for this experimental model is unknown, as well as their possible coordinated and altered expression, evidence that could to expand our knowledge about melanoma plasticity is seen from the microRNAs post-transcriptional regulation mechanisms, and at least in part, to explain the changes in gene expression previously reported for this model of proliferation reduction and differential pigmentation.

Our objective was to assess miRNA expression changes and examine whether a coordinated expression of miRNAs and their potential targets are associated with the differential pigmentation and reduction of proliferation in B16F1 melanoma cells exposed to 5-Brd-2′-dU and L-Tyr. This work contributes to the knowledge about specific microRNA profiles in B16 melanoma cells under growth suppression and differential pigmentation. L-Tyr and 5-Brd-2′-dU induced coordinated expression on potential molecular targets and enhance the hypothesis of microRNAs that affect cellular programs by exerting control over different genes.

## 2. Results

### 2.1. Decreased Proliferation and Pigmentation Changes in Melanoma B16F1 Cells

The MTT assay and the exclusion of Trypan Blue revealed a reduction in cell B16F1 melanoma cells. Decrease in viable cells (to less than 50%) secondary to 72 h-long treatment with 5.0 mM amino acid L-Tyrosine (L-Tyr) (*n* = 3) or 2.5 μg/mL thymidine analog 5-Bromo-2′ deoxyuridine (5-Brd-2′-dU) (*n* = 3); in both cases, exposure for 72 h to L-Tyr and 5-Brd-2′-dU generated a statistically significant decrease in the number of B16F1 cells, from 3.6 × 10^6^ ± 1.16 × 10^5^ to 7.4 × 10^5^ ± 9.23 × 10^4^ (79% reduction) and 1.3 × 10^6^ ± 5.5 × 10^4^ (64% reduction), respectively (Figure 1b; Appendix A). Cells exposed to 5-Brd-2′-dU showed more expanded and flattened forms, while cells exposed to L-Tyr presented morphology similar to that of melanocytes with the presence of longer dendritic processes (Figure 1a). We observed these changes over time (240 h) (Appendix A). Morphological changes and cell proliferation changes had already been reported previously for exposure to L-Tyr [19,20] and 5-Brd-2′-dU [21,22], although there were variations in exposure concentrations.

Reduction in cell number was associated with death. The number of cells in supernatants by I.P. incorporation; in unexposed cells (Control), the value was 1.1 × 10^4^ ± 1.1 × 10^3^, while for L-Tyr exposure, it was 2.9 × 10^4^ ± 4.8 × 10^3^. For 5-Brd-2′-dU, it was 5.1 × 10^4^ ± 1.1 × 10^4^. The values obtained in supernatants were on average (97 X) and (45 X) lower than the differences found by trypan blue in unexposed B16F1 cells (Control) and its counterpart, cells exposed to L-Tyr and 5-Brd-2′-dU. The above suggests that other mechanisms may explain the reduction of cells; therefore, we calculated the population doubling times from the MTT reductase activity assay and its corresponding calibration curve (Figure 1d; Appendix A). The population doubling times increased from 19.6 ± 3.94 h (CV, coefficient of variation of 20%) to 48.67 ± 6.25 h (CV of 13%) and 27.03 ± 3.0 h (CV of 11%) for exposure to L-Tyr and 5-Brd-2′-dU, respectively. These differences indicated that reducing the number of cells at 72 h would be the consequence of variations in the cell cycle control.

Cell cycle analysis (Figure 1e) showed changes in the DNA content of cells exposed to L-Tyr and 5-Brd-2′-dU. Indeed; statistically, significant changes occurred in the transition of the G0/G1 phase (from 50 ± 3.0% to 66.6 ± 2.8%) and G2/M (from 15.7 ± 1.6% to 2.4 ± 1.2%) for 5-Brd-2′-dU and in the S phase for L-Tyr (from 24.4 ± 0.7% to 15.8 ± 3.6%). The cells exposed to L-Tyr significantly increased in the G0/G1 phase (55.5 ± 3.8%, *n* = 3).

Based on the method described [23,24], we quantified the formation of melanin pigment in culture B16F1 melanoma cells under standardized conditions [15,25] and 5-Brd-2′-dU [14,21,26]. Synthetic melanin helped construct calibration curves to quantify by spectrophotometry and fluorometry (Appendix A). The quantification of melanin by both methods was comparable (Figure 1b) (Pearson’s correlation coefficient 0.999). The two approaches allowed the detection of increase (9 X) and decrease (5 X) in melanin concentration in B16 cells exposed to L-Tyr and 5-Brd-2′-dU, respectively (Figure 1f). However, fluorometry was more sensitive (detection limit 0.098 μg/mL and CV of 9.85%). Up to this point, the pooled results confirmed that exposure to L-Tyr and 5-Brd-2′-dU induces an arrest at different points in the cell cycle control for B16F1 cells and differential pigmentation phenotype that could be associated with changes in the expression of genes involved in melanin synthesis.

### 2.2. Induction of Replicative Senescence in B16F1 Cells Exposed to 5-Brd-2′-dU

5-Brd-2′-dU induced replicative senescence, a phenotype described in different cell models [27,28,29], including B16 cells [30,31]. We confirmed by Senescence-associated beta-galactosidase (S.A. β-Gal) assay, histone H3K9me3 labeling, apparent cell area measurement and flow cytometry, size analysis (FSC), and complexity (SSC) (Figure 2). Although not all cells were positive at pH 6.0 (Figure 2a), the chromogenic S.A. β-Gal assay revealed a significant increase (3.7 X) in the number of cells positive to the substrate X-Gal by exposure to 5-Brd-2′-dU against control or those exposed to L-Tyr (Figure 2b). 

The histone H3K9me3 associated with heterochromatin’s remodeling ([31,32] increased in cells exposed to 5-Brd-2′-dU revealed by higher fluorescence intensity. The number of concentrated foci, which is associated with heterochromatin (SAHF) [32], some of these SAHF, coincides with a greater density obtained by DAPI labeling (Figure 2c).

Although the cells exposed to both stimuli showed greater apparent cell area (2.6 × 10^−2^ mm^2^ for L-Tyr and 3.3 × 10^−2^ mm^2^ for 5-Brd-2′-dU) compared to the control (1.5 × 10^−2^ mm^2^) (Figure 2d); cells with 5-Brd-2′-dU changed in size (FSC, quadrant Q3) going from 3.8% to 9.7%. These cells also had higher metabolic activity measured by MTT reductase assay (Appendix A), where the slope of the calibration curve was 1.39 compared to 1.22 for the control. Overall, the results of exposure to 5-Brd-2′-dU suggest a phenotype associated with senescence and agree with that previously reported in 5-Brd-2′-dU-induced senescence models, where an increase in lysosomal organelles would explain the more significant S.A. β-Gal activity and increase in messenger RNA [33,34,35]. Exposure to L-Tyr, the cells increased both in complexity (SSC) and in size (quadrants Q1 and Q2), going from 5.4 and 3.3% in the control cells to 26.7 and 20.8%, respectively (Figure 2e).

### 2.3. Small RNAseq and Differential Expression Analysis of miRNAs

The alteration of different cellular programs in melanoma, control of the cell cycle, pigmentation, and senescence are associated with changes in the expression of miRNAs and their effect on the post-transcriptional regulation of target genes [11,36]. For the phenotypes induced in B16F1 cells, exposure to L-Tyr or 5-Brd-2′-dU causes changes in hundreds of genes [14,31,37]. However, it is unknown whether there are differences in the expression of miRNAs in these phenotypes. For this reason, we proceeded to identify and quantify by small RNAseq the expression of miRNAs in cells exposed to 5-Brd-2′-dU and L-Tyr (Appendix A) and establish the differences in expression concerning non-exposed cells (Figure 3).

The quality and integrity number (RIN) of the extracted RNA was within the values required for the Small RNAseq (Appendix A). The average of readings obtained after sequencing for each experimental group (*n* = 3) was 1.38 × 10^7^, 8.0 × 10^6^, and 1.03 × 10^7^ for control cells, exposed to L-Tyr or 5-Brd-2′-dU, respectively (Appendix A); these readings presented Phred score values close to 40, indicating a 99.99% precision in the database call (data not shown, obtained by FastQC). With the readings, we constructed an expression matrix that contained in each row the identification number (I.D.) of the mature miRNA for sequences that presented 100% agreement with the sequences reported in miRBase version 21. We listed on each column the total number of counts for each miRNA. The principal component analysis (PCA) showed greater homogeneity in the control samples and a more significant variation in those exposed to 5-Brd-2′-dU. When running DESEq2 on the expression matrix’s normalized data, some miRNAs changed, measured as the Log_2_ Fold Change, which suggests differential expression (M.A. plot of Appendix A), some of them, when correcting multiple tests, maintained their statistical significance (Table 1). In cells exposed to L-Tyr, 35, 21 were under-expressed, and 14 over-expressed. In cells exposed to 5-Brd-2′-dU, the number of miRNAs with differential expression was 32, 11 of them under-expressed, and 21 over-expressed, all concerning control B16F1 cells (Table 1).

Similarities and differences in the expression of the miRNAs obtained in cells exposed to L-Tyr or 5-Brd-2′-dU concerning the control were shown with heat maps. We exclusively included the list of differentially expressed miRNAs (Figure 3a; Appendix A). As shown in Figure 3a the patterns obtained were common in overexpression for miR-30d-5p, miR-129-5p, miR-99b-5p, and miR-191-5p and of under-expression for members of the *let*-7 family (a-5p, d-5p), and the miRNAs, miR-23a-3p and miR-361-5p. The miR-211-5p was the only miRNA that presented overexpression in cells exposed to L-Tyr and under-expression with 5-Brd-2′-dU. However, the comparison of miRNAs differentially expressed in cells exposed to 5-Brd-2′-dU concerning L-Tyr yielded another set of miRNAs with orders in differential expression not visualized under the previous comparison exposure model the control (Table 1 and Appendix A). Since this case’s expression level was not the same, the exposure to L-Tyr or 5-Brd-2′-dU may generate changes in some miRNAs’ expression in a differential way. (Appendix A). Overall, the miRNAs’ expression revealed a different expression profile in B16 cells in response to exposure to L-Tyr or 5-Brd-2′-dU. This profile was related to the observed differential pigmentation, proliferation, and senescence changes induced in this cellular model.

### 2.4. Confirmation of miRNAs Differential Expression by RT-qPCR Stem-Loop and Prediction of miRNA Targets Associated with Induced Phenotypes

We confirmed in terms relative differential expressions (rER) the expression of 7 miRNAs reported in Table 1 (211-5p, 30d-5p, 148b-3p, 129-5p, 27b-3p, 470-3p and 470-5p) by RT-qPCR stem-loop (Figure 4) (*n* = 3). miR-211-5p, increased 2.5 X by exposure to L-Tyr and a decreased 2 X, for 5-Brd-2′-dU (Figure 4a; Table 1). On the other hand, the overexpression of miR-30d-5p, increased by 5.4 X and 8 X (Figure 4c; Table 1) and of miR-129-5p, increased by 4.5 X and 1.4 X (Figure 4c; Table 1) for exposure to L-Tyr and 5-Brd-2′-dU, respectively (Figure 4c; Table 1). RT-qPCR found miR-27b-3p, miR-148b-3p, a differentially expressed with values that did not agree in trend to those reported in Table 1 in DESEq2. Compared to the reduction registered by small RNAseq due to exposure to L-Tyr (miR-27b-3p 1.5 X and miR-148b-3p 2.0 X), by qPCR, both miRNAs increased their expression concerning the control (3.1 X and 4.5 X) respectively (Figure 4c; Table 1). For exposure to 5-Brd-2′-dU, we found an increase (3.3 X and 3.55 X) in expression by qPCR.

miRNAs showed common and coordinated expression patterns in this differential pigmentation and reduced proliferation induced in melanoma B16F1 cells. We identified potential targets of miRNAs by prediction with TargetScanMouse (Figure 4a–c). Some genes participated in controlling the cell cycle and senescence (i.e., *Ccnd1*, *Cdk2*, *Cdk4*, and *p21*/*Cdkn1a*, *Rb1*, *E2F2*, *Atm*, *Foxm1*, and *E2F3*). Other targets (i.e., *Mitf*, *Tyr*, *Tyrp1*, or *Dct*) were involved in melanin synthesis. miRNAs presented changes in their expression and targeted genes associated with controlling the cell cycle and senescence as expected (Figure 4a, Table 1).

### 2.5. Regulatory Networks and Co-Expression Networks (RC-miR)

The miRNAs are associated with melanoma’s cellular processes independently; however, we intend to relate the effect they would have together and their potential molecular targets on the phenotype induced by L-Tyr y 5-Brd-2′-dU. We constructed possible gene interaction networks using the miRNet web application with the differentially expressed miRNAs, their putative genes, and functional enrichment analysis by KEGG signaling pathways (Figure 3b–d, Appendix A Functional Enrichment using KEGG).

In cells exposed to L-Tyr, enrichment revealed a total of 269 KEGG pathways, 2641 hits, and 1458 nodes; of which, 36 routes with 828 hits passed the hypergeometric test (Figure 3b). In cells exposed to 5-Brd-2′-dU, 256 KEGG pathways, 1560 hits, and 962 nodes; 11 routes and 169 hits with statistical significance (Figure 3c). Of these pathways, the regulation of proliferation, apoptosis, differentiation, survival, and pathways associated with cellular metabolism related to the phenotype induced by exposure to L-Tyr; while the p53, FoxO, apoptosis, Jak-STAT, and altered transcriptional regulation in cancer pathways related to the phenotype caused by exposure to 5-Brd-2′-dU.

We reviewed the gene regulation network constructed for the miRNAs differentially expressed in cells exposed to 5-Brd-2′-dU concerning L-Tyr. The network contained a total of 255 KEGG routes, 1671 hits, and 903 nodes; of which, 21 routes with 337 hits passed the hypergeometric test (Figure 3d). In this case, we observed other pathways involved, such as cell cycle, Wnt, and ubiquitin-mediated proteolysis. The data reported so far show the existence of a specific miRNA expression profile for each induced phenotype. This profile, seen from a network gene regulation model, would be associated with controlling different cellular programs through their target genes.

We used mutual information to measure similarity and co-expression networks (RC-miR) based on adjacency matrices for the differentially expressed miRNAs. As can be seen in Figure 4d, in cells exposed to 5-Brd-2′-dU, 9 clusters of miRNAs were identified, two of them over-expressed (miR-129-3p, miR-99b-5p, miR-378-3p, and miR-30d-5p) and (miR-129-5p, miR-21a-5p, and miR-335-3p), which according to TargetScan would have as a common target the factor *E2F2*; the cluster miR-455-3p, miR-29c-3p and miR-335-5p on *p21* and the cluster miR-29c-3p and let-7e-5p on *Ccnd1* (Figure 4a). While for under-expressed miRNAs, we found the cluster let-7a-5p and let-7b-5p that would have as a putative target *Tyr*, *Ccnd1*, *Cdk4*, *E2F2*, *E2F3*, and *Rb1*.

In cells exposed to L-Tyr, 10 sets were found, of which, it was coincident with 5-Brd-2′-dU, finding over-expressed miR-129-3p, miR-99b-5p, and miR-30d-5p and miR-211-3p, miR-149-5p and miR-129-5p clusters as a possible target of *p21*. We found two clusters of potential regulators for *Ccnd1*: one with over-expressed miRNAs (miR-149-5p and miR-129-5p) and one under-expressed (miR-320-3p and miR-23b-3p), which could imply a coordinated regulation by expression to the high and low of the whole set of miRNAs. Regarding sets of under-expressed miRNAs, we found a cluster with the *let*-7 family (a, c, d) and the miR-151-5p, whose potential target was *E2F2*.

### 2.6. Analysis of Differential Expression mRNAs and Protein Level

The mRNA expression levels of the selected genes were quantified by RT-qPCR, which we reported as the relative of differential expression (rER) concerning the housekeeping gene GAPDH and B16F1 cells. Western blot and immunofluorescence determined the expression and location of the corresponding protein. We evaluated some mRNAs involved in the cell cycle and senescence: Cyclin D1 (Ccdn1) expression, cyclin-dependent kinases Cdk4 and Cdk2, and the cyclin-dependent kinase inhibitor p21 (Figure 5). We evaluated the genes Tyr, Tyrp1, and Dct associated with melanogenesis and the master regulator of cell differentiation Mitf (Figure 6) for pigmentation phenotype. The relative expression (rER) of Ccnd1 decreased for L-Tyr by 0.47 ± 0.16 (2.5 X). Exposure to 5-Brd-2′-dU induced a decrease of 1.6 X (0.6 ± 0.06) (Figure 5a). 

The Cdk4 mRNA showed a decrease concerning the control cells only for L-Tyr with rER of 0.79 ± 0.1 (1.25 X) (Figure 5d). As is known, the sustained activation of the binding of the CCDN1 protein to the CDK4 protein kinase allows the early transition from G1 to S in the cell cycle, increasing the phosphorylation of the Rb retinoblastoma protein and releasing the E2F family of transcriptional regulators [38,39]. In both cases, the decrease in *Ccnd1* was consistent with the increase in the number of cells arrested in G0/G1 observed in cells exposed to 5-Brd-2′-dU (Figure 1e). Although in *Cdk4* it only presented a statistically significant reduction for exposure to L-Tyr, in both cases, the decrease in these mRNAs coincides with the increase in miR-129-5p and 30d-5p (Figure 4d), which as a predictor have *Ccdn1* and *Cdk4* as molecular targets (Figure 4a).

We observed a reduction in the *Cdk2* in cells exposed to 5-Brd-2′-dU (0.36 ± 0.09) (2.7 X) and non-significant variation at the mRNA level (1.09 ± 0.09) (Figure 5g); however, at the protein level, we found changes for cells exposed to L-Tyr (Figure 5h,i). A set of miRNAs with regulatory potential for each type of treatment from the coincidences reported for *Ccdn1* and *Cdk4*, except for miR-149-5p.

We founded the decrease in proliferation induced by L-Tyr and 5-Brd-2′-dU correlated with an increase in *p21* (rER 2.7 ± 0.4) (2.7 X) for cells exposed to L-Tyr and an rER of 0.67 ± 0.1 (1.5 X) in cells exposed to 5-Brd-2′-dU (Figure 5j). This increase in *p21* would explain, at least in part, the decrease in the S phase for cells exposed to L-Tyr and *CdK2* reduction with the arrest in G2/M (Figure 1e). The p21 expression increased after 7 days, maintaining the exposure (Figure 5m). In this case, the early regulation of the cell cycle by sets of miRNAs presents differential expression. They regulate other targets, such as p27 (increased in western for exposure to 5-Brd-2′-dU, data not shown).

Further, the exposure of B16F1 cells to 5-Brd-2′-dU generated a 4 X decrease in the mRNA of all proteins related to melanogenesis, with rER values of 0.24 ± 0.069 for *Mitf*, 0.29 ± 0.03 for *Tyr*, 0.29 ± 0.03 for *Tyrp1*, and in the case of *Dct*, 0.27 ± 0.05 (Figure 6a,d,j, respectively); additionally, this reduction related to the protein products obtained in each case (Figure 6b,c,e,f,h–k). These results coincide with the drastic reduction in melanin observed in cells exposed to 5-Brd-2′-dU (Figure 1f)) [14,26,35]. We did not find significant variations in the expression of Mitf mRNA (1.12 ± 0.4) (Figure 6a) and Tyrp1 (0.85 ± 0.09) (Figure 6g), but a significant increase for Tyr (1.5 ± 0.3) (Figure 6d), and Dct (1.12 ± 0.07) (Figure 6g).

## 3. Discussion

The number of reports related to changes in miRNAs’ expression and their targets in melanoma samples under different tumor progression stages is growing; however, their role in growth suppression models and whether these miRNAs change reciprocally with the phenotype are little known. We present the first report about changes in miRNAs’ expression profile in mouse B16 melanoma cells under growth suppression and differential pigmentation induced by exposure to the amino acid L-Tyrosine or the thymidine analog 5-Bromo-2′-deoxyuridine. We propose some network models of co-expression of microRNAs that present potential common targets associated with cell cycle control, senescence, proliferation, and melanogenesis in the mouse melanoma B16-F1 cells.

Exposure for 72 h to L-Tyr and 5-Brd-2′-dU generates a statistically significant decrease in the number of B16F1 cells (79% and 64%, respectively (Figure 1b and Appendix A)), results that agree with previous reports on B16 cells and other models of melanoma, such as embryonic retinal pigment cells and IIB-MEL-J [15,19,20,21,22,40]. To discriminate the loss of cells by death and cell cycle arrest, we quantified the number of cells in supernatants by I.P. incorporation, and we calculated the population doubling times (Figure 1d; Appendix A). The increase from 19.6 ± 3.94 h to 48.67 ± 6.25 h and 27.03 ± 3.0 h induced by 72-h exposure to L-Tyr and 5-Brd-2′-dU, respectively, indicates that reduction of the number of cells would be mainly a consequence of variations in the cell cycle control. Cell cycle analysis (Figure 1e), indicating a transition of the G0/G1 phase (from 50 ± 3.0% to 66.6 ± 2.8%) and G2/M (from 15.7 ± 1.6% to 2.4 ± 1.2%) for 5-Brd-2′-dU and in the S phase for L-Tyr (from 24.4 ± 0.7% to 15.8 ± 3.6%). The cells exposed to L-Tyr significantly increase the G0/G1 phase (55.5 ± 3.8% *n* = 3).

p16 is mutated in B16 cells [41] (not shown data); therefore, we decided to evaluate the expression of *p21* to understand the decrease in proliferation induced by L-Tyr and 5-Brd-2′-dU. Cells exposed to L-Tyr increase the *p21* expression with an rER 2.7 ± 0.4 (2.7 X), and cells exposed to 5-Brd-2′-dU decrease *p21* with an rER of 0.67 ± 0.1 (1.5 X) (Figure 5j). This increase in *p21* would explain, at least in part, the decrease in the S phase for cells exposed to L-Tyr (Figure 1e), while the reduction of p21 is related to the arrest in G2/M in *CdK2* in 5-Brd-2′-dU (Figure 1e) [42,43,44,45]. CDK2 is a cyclin-dependent kinase that participates in the late transition of G1/S by binding to cyclin E [46] and in the passage of G2/M by binding to cyclin A [47]. The late participation of p21 in regulating the cell cycle under exposure to 5-Brd-2′-dU [27] agrees with its high expression after seven days of maintaining the treatment (Figure 5m).

B16F1 melanoma cells under standardized cell culture conditions [15,25] and 5-Brd-2′-dU [14,21,26,35] presented changes in melanin concentrations (Figure 1b) (Pearson’s correlation coefficient 0.999). Fluorometry and colorimetry allowed the detection of increase (9 X) and decrease (5 X) in melanin concentration in B16 cells exposed to L-Tyr and 5-Brd-2′-dU, respectively (Figure 1f). However, fluorometry was more sensitive (detection limit 0.098 µg.mL^−1^ and CV of 9.85%); this hyperpigmentation has also been described in primary culture melanocytes exposed to high concentrations of the amino acid L-tyrosine [48,49].

We confirm that exposure to L-Tyr and 5-Brd-2′-dU induces an arrest at different points in the cell cycle control for B16F1 cells and differential pigmentation phenotype associated with changes in the expression of genes involved in melanin synthesis [17,31,50]. L-Tyrosine and the thymidine analog, 5-Bromo-2′-deoxyuridine induced a decrease in proliferation and pigmentation changes in melanoma B16F1 cells. These phenotypic changes remember terminal differentiation and replicative senescence induced by L-Tyr and 5-Brd-2′-dU, respectively.

5-Brd-2′-dU’s replicative senescence generates a phenotype associated with induced senescence in different cell models [27,28,29], including B16 cells [30,31,51]. Senescence-associated beta-galactosidase (S.A. β-Gal) assay (Figure 2b) and positivity for the histone H3K9me3 involved in heterochromatin remodeling confirm the senescence phenotype [52,53]. Cells exposed to 5-Brd-2′-dU show many concentrated foci, which are associated with heterochromatin (SAHF). Some of these SAHF coincide with a greater density obtained by DAPI labeling (Figure 2c) [32]. Overall, the results of exposure to 5-Brd-2′-dU suggest a phenotype associated with senescence and agree with that previously reported in 5-Brd-2′-dU-induced senescence models [33,34]. Exposure to L-Tyr induces complexity (SSC) and increases cell size (Figure 2d), perhaps due to the increased number of melanosomes required to synthesize and store the melanin produced [20,54,55].

A global change in miRNA expression profiles by exposure to L-Tyrosine generated 61% of under-expressed miRNAs and 39% over-expressed, compared to 34% and 65% 5-Brd-2′-dU. These results suggest that 5-Brd-2′-dU’s melanoma cells exposed to genotoxic or metabolic stress by L-Tyr generate two expression profiles for the same genetic background, explaining the reduction in growth and changes in the pigmentation described here, depending on the potential regulation on its molecular targets.

As expected, the results presented here contrast with previous reports of miRNA expression profiles already obtained under discrete stages of melanoma progression. For example, miR-211-5p in this experimental model is over-expressed by the exposure to L-Tyr; in contrast, during tumor progression, miR-211-5p tends to decrease [56,57,58]. For the let-7a family members, under-expressed in both L-Tyr and 5-Brd-2′-dU or let-7b overexpressed in 5-Brd-2′-dU and reported as a cell cycle regulator in melanoma, possibly through *cyclins D1* and *D3* and *Cdk4* and associated with interference with anchorage-independent growth [59]. miR-455, high in B16 cells exposed to 5-Brd-2′-dU and low in tumor progression [58,60]. These findings agree with the notion that some miRNAs are known to participate in tumor progression as a tumor suppressor, oncomiRs, or miRs and may involve a possible reversible tumor reprogramming mechanism through their targets, in processes that include cell proliferation, senescence, and melanogenesis.

5-Brd-2′-dU is a potent tumor cell radiosensitizer [61]. Its incorporation into DNA generates lesions that include mutations, fragile sites, chromatid-cracking and exchange, micronuclei, and hypermethylation [27]. DNA damage produces disruptive effects in the patterns of DNA transcription, increase the affinity of nuclear factors, altering the binding of regulatory proteins to the DNA, which may result in changes in cell differentiation, suppressing cell proliferation, and even causing alterations in the plasma membrane cells [21,22,27,33,34,61,62,63,64].

The results presented herein B16 cells on microRNA expression changes extend our knowledge on other molecular effects generated by 5-Brd-2′-dU exposure in terms of a possible regulation of these processes mediated in a post-transcriptional manner. In B16F1 melanoma cells, 5-Brd-2′-dU causes an absence of melanosomes, an interruption of the lamellar matrix of the endoplasmic reticulum, and an irregular perinuclear microfilament arrangement [18].

We observed changes at the mRNA and protein levels for Tyr and Dct and protein for Tyrp1 without *Mitf* gene expression changes, suggesting that other regulatory mechanisms are related, i.e., regulation mediated by sets of miRNAs. Downregulation of tyrosinase gene expression by 5-Brd-2′-dU is not due to incorporating this analog into upstream sequences of the tyrosinase gene. Therefore, the mechanism of pigment inhibition is not elucidated yet [21,26,65]. In contrast, in cells exposed to L-Tyr, the Mitf values did not present statistically significant changes concerning the control cells (Figure 6). As shown here, cells exposed to 5-Brd-2′-dU and a decrease in Mitf, the master regulator of melanogenesis, modulate *Tyrosinase*, *Tyrp1*, and *Dct* gene expression [20,55,66,67,68,69] and through the coordinated and simultaneous gene activity interference.

Pigmentation changes in this model are not mutually exclusive with the suppression of cell growth, which supports the idea of multiple signaling pathways and molecular networks operating differentially to generate a reduction in cell proliferation with differentially hyper or hypopigmentation phenotypes. Perhaps the latter, as a consequence of the phenotype associated with senescence due to 5-Brd-2′-dU exposure, evidence widely reported in other models where 5-Brd-2′-dU generates hypopigmentation [21,26,70].

The confirmation of differential expression of some miRNAs by RT-qPCR stem-loop and the prediction of miRNA targets associated with induced phenotypes reveals a different set of miRNAs per challenge 5-Brd-2′-dU and Tyr the differential pigmentation phenotype. However, we did not find miRNAs with differential expression by TargetScan that had *Dct* as a target and for miR-27b-3p, which has *Mitf*, *Tyr*, and *Tyrp1* as a common target and miR-148b-3p, reported as a *Mitf* regulator [71], DESEq2 and RT-qPCR found non-concordant values. In this regard, for exposure to L-Tyr, in PCR, both miRNAs increased their expression concerning the control, a situation also recorded for exposure to 5-Brd-2′-dU, a differential expression that was not obtained by sequencing (Figure 4c). This apparent miRNA antagonism is frequent when the differential expression analysis is assessed by different platforms, including microarrays, NGS, and RT-qPCR. It would possibly be associated with the technical effects of each methodology [72], for what was considered the analysis of expression by PCR as gold standard [73].

miR-470-5p targets *Mitf* that regulate Tyr enzyme, which is a target of miR-470-3p; had already been reported before by our laboratory [18]. Here we confirmed by RT-qPCR (Figure 4c), with an increase due to exposure to 5-Brd-2′-dU (22 X for miR-470-5p), but not for L-Tyr. These two miRNAs participate in the differentiation of mouse embryonic cells [74] and hepatocellular carcinoma [75]. Our results show that their potential role in the CCDN1/CDK4 complex as an essential oncogenic driver in various cancer types, including melanoma [76].

The let-7 family’s downward expression agrees with the previous publications [58,77,78] through binding to β3 integrin [79] and the miR-125a-5p/Lin28B/*let*-7 regulatory circuit [80], metastasis [81], metabolism [82], and the control of the cell cycle by CDK4 and cyclins D1, D3, and A [59].

miR-129-5p has been reported in the increase in the size of A375 cells through binding to AEG1 [83]. In other types of cancer, this miR is related to cell cycle control, proliferation, migration, and invasion [84] in multiple myeloma, glioma cells [85,86], prostate cancer [87,88], colorectal cancer [89], and breast cancer [90].

miR-30 family is related to the regulation of proliferation processes via Wnt/β-catenin [91,92], cell cycle control [93], apoptosis [94,95] and senescence in various cell models [96,97]. In melanoma, it is related to increased cell invasion due to increased aberrant glycosylations associated with its molecular target GALNT7 [98].

Pigmented and unpigmented B16 melanoma cells present an expected decrease in cell proliferation. It is possible to suggest that miRNAs share, through their potential targets, a reciprocal and coordinated regulation on the cell cycle’s control dependent on the expression level of the specific group of miRNAs, without ignoring the possible differential effect that the regulation of each of them would exert.

L-Tyr induces melanin production and increases the synthesis of proteins related to melanogenesis as previously reported [55,99], trends confirmed in B16F1 cells exposed to L-Tyr (Figure 1f). *Mitf* did not change in cells exposed to L-Tyr; therefore, the differences found at the mRNA and protein levels for TYRP1, as well as the increase in TYR and DCT suggested that, in melanogenesis, other types of post-transcriptional regulation would be involved, such as the differential expression of sets of miRNAs: miR-211-5p and miR-148b-3p, previously reported as regulators of the pigmentation. Different miRNAs regulated *Mitf* and, in turn, is a transcriptional regulator of *p21*, *Cdk2* [100,101,102] and miR-211-5p, the latter, which regulates pigmentation by targeting EDEM1, a TYR inhibitor. miR-211-5p increased to 2.5 X by exposure to L-Tyr and decreased 2 X by 5-Brd-2′-dU. We did not find a miR-211-5p direct specific gene target involved in the melanogenesis. However, this was the only miRNA that showed differences between the two phenotypes—pigmented and unpigmented melanoma cells. It is exciting that miR-211-5p has been proposed as an indirect regulator of pigmentation through binding to the TGF β-2 receptor, a factor responsible for the transcription of pigmentation-associated genes Dct, TYRP1, TYR, and PMEL17 [103] and binding to the TYR inhibitor EDEM1 mRNA [104]; additionally, MITF may regulate the expression of miR-211-5p [103]. Together, these results suggest a possible pigmentation regulatory circuit in which genes regulated by miR-211 and the MITF factor would be participating, including the coordinated expression of sets of miRNAs.

The analysis of functional enrichment by KEGG pathways for the differentially expressed miRNAs shows diametrically compromised pathways for each phenotype that reinforce the hypothesis of miRNAs that operate in the regulation of signaling pathways under metabolic stress generated by exposure to L-Tyr (PI3K/Akt, MAPK, mTOR, among others) and miRNAs with potential participation in response pathways to genotoxic stress (p53, FoxO, Jak-STAT and transcriptional alteration in cancer).

Coordinated and reciprocal miRNA expression changes may explain changes in melanoma phenotypes induced by L-Tyr and 5-Brd-2′-dU. miR-211-5p is overexpressed in the hyperpigmented phenotype and downward in the hypopigmented phenotype, without ignoring other miRNAs’ possible participation such as miR-470-5p miR-148-3p, miR-30d-5p, and miR-27b-3p, which have *Mitf*, *Tyr*, and *Tyrp1* as potential molecular targets. It was striking that exposure to L-Tyr did not affect the expression levels of the gene and protein of MITF, the master regulator of the expression of TYR, TYRP1, and DCT, but for the levels of *Tyr* mRNA, an increase with statistical significance. These results suggest an additional regulation to the transcription factor, possibly involving miR-211-5p, as has already been reported in other models [104].

Regarding the control of the cell cycle and senescence in B16F1 melanoma cells, where p16INK4a is truncated [41], suggests other pathways involved, such as p21, p53 pathway, cyclin D1, Cdk2, Cdk4, Rb, and transcription factors E2F [105,106,107]. 

This bioinformatics approach based on experimental data of differential expression of miRNAs could facilitate the establishment of new hypotheses associated with a differential expression profile, whose expression as a whole could be related to the synchronous regulation of gene expression in particular cellular programs. Recent evidence suggests a possible epigenetic reprogramming by transcriptional modification of miRNA expression profiles [108]. Based on the results in this experimental model of induced growth suppression and differential pigmentation, cells may reprogram a reversible phenotype through the coordinated and reciprocal regulation of their transcriptome. It is tempting to speculate a possible coordinated response under the stimulus of genotoxic or metabolic stress described above. This response involves the global reprogramming of patterns of miRNA expression and miRNAs relative antagonism.

A parallel reprogramming in mRNA expression may provide a partial explanation for the changes reported in cell cycle control, senescence, and melanogenesis; new studies are necessary to understand the molecular bases of miRNAs and epigenetic regulation. However, it is necessary to confirm the clusters generated here in the network models and establish the functional network associations between these microRNAs and their mRNAs target in cell cycle control and pigmentation. 

The results apply only to the murine melanoma cell line B16 under specific experimental conditions of this study. They do not allow for any generalization and cannot extrapolate the results found in our work to another melanoma model given the high biological variability. In this regard, we compared the same population of the cell line under two different phenotypes to have the same genetic background to decrease biological variability. While a basis for comparison is still maintained, many differences between cell lines and primary cell cultures and the heterogeneity in melanoma from biopsies or surgical specimens may limit these findings’ relevance to the miRNA expression. They may be different and questionable for the genetic variations. Therefore, it is necessary to confirm the findings in other melanoma cells, especially in primary human melanoma cultures.

## 4. Materials and Methods

### 4.1. Cell Line Cultures

The B16F1 tumor cell line: (ATCC^®^ CRL-6322™, Manassas, VA, USA), with epithelial morphology and adherent growth, from C57BL/6J mouse skin, was cultivated at 37 °C with 5% CO_2_ and 98% humidity (Precision Scientific NAPCO 5410-120 Water Jacketed CO_2_ Incubator, Buffalo, NY, USA) in DMEM advance (Dulbecco’s modification of Eagles medium) (Gibco, Thermo Fisher Scientific, Waltham, MS, USA) supplemented with 10% (*v*/*v*) fetal bovine serum (SFB) (Gibco), pH 7.2–7.4 condition, penicillin (100 U/mL), and streptomycin (100 µg/mL). The cells, at 70% confluence, treated with trypsin-EDTA (Sigma-Aldrich, St. Louis, MO, USA) 0.25% (*m*/*v*), resuspended and counted in automatic Tali™ Image-based Cytometer (Thermo Fisher Scientific) following the manufacturer’s specifications and their viability evaluated in Newbauer’s Chamber in dilution 1:1 with Trypan Blue (Sigma-Aldrich).

### 4.2. Induction of the Phenotype of Decreased Proliferation and Changes in Pigmentation by Expo-Sure to L-Tyrosine (L-Tyr) or 5-Bromo-2′-Deoxyuridine (5-Brd-2′-dU)

We seeded an average of 1.5 × 10^5^ B16F1 cells in complete DMEM in a sterile Petri dish (Falcon 3003). Between 10 to 12 h after sowing, we changed by fresh medium supplemented with 5-Brd-2′-dU (Sigma-Aldrich) or L-Tyr (Biomedical Inc, Clinton Township, MI, USA) at a final concentration of 2.5 µg/mL and 5 mM, respectively, and incubate for 72 h without medium change. Furthermore, we followed every 48 h for cell monitoring assay until 240 h, where 5.0 × 10^2^ cells were seeded to determine population doubling times. The photomicrographs were captured under a 20x inverted light microscope (Nikon Eclipse Ti, Kobe, Japan) using the NIS-Elements-Nikon program. The cells treated with 5-Brd-2′-dU, protected from light, were resuspended in complete medium, and the exclusion of Trypan Blue assessed the viability. Additionally, the cells from the culture supernatants recovered by centrifugation, and their viability was determined using propidium iodide (I.P.) incorporation assay in Tali™ Image-based Cytometer following the manufacturer’s specifications.

### 4.3. MTT Assay

B16F1 cells exposed or not to L-Tyr or 5-Brd-2′-dU were seeded in a sterile 96-well culture box (Corning, NY, USA); after 10-12 h of incubation, the medium was removed and incubated again for 1h at 37 °C with 0.83 mg.mL^−1^ of MTT (3 (4.5 dimethylthiazol) 2.5 diphenyl tetrazolium bromide) (Sigma-Aldrich) dissolved in phosphate-buffered saline (PBS). We removed the MTT excess MTT, and dimethyl sulfoxide (DMSO) (Merck, Branchburg, NJ, USA) to help to dissolve the formazan crystals. The wells were read at 560 nm on a GloMax^®^-Multi Detection System (Promega, Madison, WI, USA) following the manufacturer’s specifications. To determine the diluted formazan crystals’ proportionality with the number of viable cells, we constructed a calibration curve using six serials 1:2 dilutions from 8.0 * 10^4^ cells. Each well-read read under the same conditions. We assessed population doubling time following the reported protocol [109].

### 4.4. Univariate Analysis of Cell Cycle by Incorporation of Propidium Iodide

For the cell cycle analysis, we followed the previously reported protocol [110]. In brief, a total of 1.0 * 10^6^ B16F1 cells exposed or not to L-Tyr or 5-Brd-2′-dU were fixed in 75% (*v*/*v*) ethanol for 2 h at 4 °C. Excess ethanol was removed and washed with PBS, and the cell pellet was resuspended in propidium iodide (P.I.) staining solution containing: 0.1% (*v*/*v*) Triton X-100 (Sigma-Aldrich); 10 µg.mL^−1^ of propidium iodide (Sigma-Aldrich) and 100 µg/mL of DNase-free RNase A (Thermo Fisher Scientific) in PBS, protected from light for 30 min at T ° room. The cells were analyzed at a flow cytometer (BD FACSAria II ™, Franklin Lakes, NJ, USA). The fluorescence was measured at 536 nm and 617 nm for excitation and emission, respectively, a total of 1.0 × 10^4^ events in three independent replicates. We estimated the relative number of cells using the FlowJo™ program that assumes Gaussian distributions of the 2N and 4N populations and using a subtractive function to establish the S phase population. Additionally, the cytometer classified the cells according to their size (FSC-A) and its graininess or morphological complexity (SSC-A). Analysis of statistical significance for multiple comparisons without correction, with an alpha = 0.05.

### 4.5. Melanin Quantification

Melanin was quantified by spectrophotometry and fluorometry following previously reported protocols [23,24]. In short, we dissolved synthetic melanin (Sigma-Aldrich) 100 µg/mL and melanin from B16F1 cells with 1 M NaOH (Merck) at 80 °C, the collected supernatants after centrifugation (1 min at 12,000× *g* at 4 °C), and we assess the optical density (O.D.) by spectrophotometry at 405 nm. For fluorescence melanin quantification, after incubation with 15% (*m*/*v*) hydrogen peroxide (H_2_O_2_) (Sigma-Aldrich), we read the supernatants on the Glomax equipment with a wavelength of 470 nm and 550 nm excitation and emission, respectively. We described the results according to the number of viable cells after determining cell number by MTT reductase assay. Data were expressed as the average of three independent trials ± standard deviation and analyzed by unpaired *t*-test with Welch correction and a 95% confidence interval.

### 4.6. Β-Galactosidase Activity Associated with Senescence (S.A. β-Gal)

To quantify S.A. β-gal activity, we used a previously reported chromogenic assay [111]. In brief, we fixed B16F1 cells with a 0.2% (*m*/*v*) paraformaldehyde solution (Merck) and 0.2% (*v*/*v*) glutaraldehyde (Sigma-Aldrich) for 10 min at 4 °C. We incubated for 18 h at 37 °C without CO_2_ with the substrate and chromogenic solution containing: 1 mg/mL of X-gal reagent (Invitrogen, Carlsbad, CA, USA), 61 mM citric acid/sodium phosphate (pH 4.0, pH 6.0 and pH 7.5; as appropriate), 5 mM K_3_Fe [CN]_6_ (Sigma-Aldrich), 5 mM K_4_Fe [CN]_6_ (Sigma-Aldrich), 150 mM NaCl (Merck) and 2 mM MgCl_2_ (Merck). We documented the changes through a photographic record and spectrophotometric quantification. The staining solution was removed and washed with PBS pH 7.4 at room temperature and protected from light. We dissolved the β-galactosidase product with 200 µL of 99% DMSO by 5 h and then quantified at 600 nm in the Glomax spectrophotometer. Data were expressed as the average of four independent trials ± standard deviation and analyzed by unpaired *t*-test with Welch correction and a 95% confidence interval.

### 4.7. Protein Detection by Immunofluorescence

Cells seeded in coverslips in the presence or absence of 5 mM L-Tyr or 2.5 µg/mL 5-Brd-2′-dU were fixed with 4% (*m*/*v*) paraformaldehyde in PBS, permeabilized with 0.3% (*v*/*v*) Triton X-100 and incubated with the rabbit polyclonal primary antibody (Santa Cruz Biotechnology Inc, Dallas, TX, USA): anti-Cdk2 (sc-163), anti-Cdk4 (sc-260), anti-cyclin D1 (sc-717), anti-p21 (sc-756), anti-TRP1 (sc-25543), anti-Dct (sc-25544), anti-Tyrosinase-Tyr (sc-15341) or anti-MITF (sc-11002), the latter made in goat. To detect H3K9me3, we used a polyclonal antibody made in rabbits (ab8898) (Abcam, Cambridge, UK), in all cases, final concentration of primary antibody was 0.4 µg/mL. As a secondary antibody, we used a CFL-647-labeled goat anti-rabbit (sc-362292) (Santa Cruz), and for H3K9me3 and MITF, a CFL-488-labeled goat anti-rabbit (sc-362262) and a chicken-marked anti-goat marked with Texas Red (sc-3923), respectively, both from (Santa Cruz) with concentrations and incubation times following manufacturer’s recommendations (0.4 µg/mL). After nuclei labeled with DAPI (4′,6-diamidino-2-phenylindole) (Sigma-Aldrich) and the coverslips mounted on 25% (*v*/*v*) glycerol (Merck), we acquired the images under an inverted microscope (Eclipse Ti, Kobe, Japan) in a bright field and with the filter corresponding to the used fluorophores. We quantified the mean fluorescence intensities (MFI) and calculated the apparent cell area from the bright-field photographs after defining each cell’s perimeter on the NIS-Elements program. 

### 4.8. Protein Expression by Western Blot

B16F1 cells were washed with sterile PBS and mechanically lysed in the presence of RIPA buffer (Sigma-Aldrich) containing 1mM phenyl-methyl sulfonyl-fluoride (PMFS) (Sigma-Aldrich) as a protease inhibitor. The samples were centrifuged at 13,000× *g* for 15 min at 4 °C. The supernatant and the protein concentration were quantified using a calibration curve constructed with bovine serum albumin and bicinchoninic acid (Pierce ™ BCA Protein Assay Kitt, Thermo Fisher Scientific) following the manufacturer’s recommendations. A total of 30 µg of protein quantified per sample was removed by 10% SDS-PAGE electrophoresis under denaturing conditions. The obtained gels were electro-transferred to a PVDF membrane (Millipore-Merck) using the Novex^®^ semi-dry system (Thermo Fisher Scientific) following the manufacturer’s recommendations. For immuno-detection, the non-specific sites of the PVDF membrane were blocked with a 1% (*m*/*v*) solution of polyvinylpyrrolidone (PVP-40) (Sigma-Aldrich) in PBS-Tween 20 (Sigma-Aldrich) [112] and incubated with the primary antibodies used in the immunofluorescence, in addition to p16INK4a/CDKN2A (ab108349) from Abcam and p27 (sc-528) (Santa Cruz). Subsequently, the incubation with the corresponding secondary antibody marked with horseradish peroxidase-HRP (VECTOR, Burlingame, CA, USA): anti-goat made in a horse (PI-9500) and anti-rabbit made in goat (P1-1000). As load control, we used the detection of the nuclear protein Lamina B1 (sc-6216) (Santa Cruz). Finally, the detection was completed by chemiluminescence using ECL-western blotting system (Amersham, Boston, MA, USA) and densitometric analysis in Fiji-ImageJ (https://fiji.sc/)

### 4.9. Extraction of Total RNA and Enrichment of Small RNAs

We followed the organic phase RNA extraction as the previously described protocol [113]. In brief, B16F1 cells were lysed in the presence of TRIzol (Invitrogen) to solubilize the melanin. The lysate was collected in 1.5 mL Eppendorf tubes and heated for 2 min at 65 °C [114]. By successive centrifugations with chloroform (Sigma-Aldrich), isopropanol (Biomedical Inc.), and 75% ethanol (Merck), the pellet dried at room temperature for 10 min and was resuspended in H2O DEPC (diethyl pirocarbonate, Sigma-Aldrich) for subsequent RT-qPCR assays and quantified by spectrophotometry (Nanodrop 2000, Thermo Fisher Scientific); or resuspended in transport buffer following the manufacturer’s recommendations, for subsequent sequencing. To obtain enriched fractions of small RNAs ≤ 200 nt, miRVana™ miRNA Isolation Kit (Ambion, Austin, TX, USA) was used following the manufacturer’s recommendations.

### 4.10. Small RNAseq

We used Illumina technology (Illumina, San Diego, California, USA) to sequence small RNAs. Before the library’s construction, evaluate the total RNA’s quality and integrity on an Agilent RNA ScreenTape System (Illumina) under RNA integrity number (RIN) and TruSeq Small RNA Library Prep Kit (Illumina), following the manufacturer’s recommendations. A total RNA fraction coupled to 5′ and 3′ adapters to the corresponding RT-PCR for 15 cycles. The amplification products fractionated by 6% polyacrylamide gel electrophoresis, and the amplified library region containing the fraction of miRNAs (140–150 nt), was separated, precipitated with ethanol, and quantified on nanodrop. The library was run on a HiSeq 2500 (Illumina) in a fast read-per-cycle 1 × 50 format (single read), averaging 1.0 × 10^7^ readings per sample. The quality control and the normalization of the readings obtained using the BaseSpace^®^ App, Small RNA v1.0 (Illumina), the sequences converted into FastQ files, and the quality of the readings was analyzed multiQC and confirmed in Galaxy by FastQC (https://usegalaxy.org/). The alignments for readings obtained between 17 and 35 nt against the reference genome for Mus musculus (mm9) and other databases including miRBase version 21 to identify mature miRNAs (http://www.mirbase.org/) in whose case was only assigned the name of the miRNAs when we found 100% identity between the reading and the miRBase reference. We registered the data at the Gene Expression Omnibus-GEO-NCBI with the access number GSE147170 (https://www.ncbi.nlm.nih.gov/geo/query/acc.cgi?acc=GSE147170).

### 4.11. Principal Components Analysis-PCA

We assessed the PCA in R code under the “plot PCA” function. To summarize the systematic patterns of variation in the readings obtained from the sequencing in principal components, we achieved orthogonal transformation by reducing each sample at one point [115], provided separate samples by expression variation, and identified possible outliers for each sample. For the construction of the 3D PCA, we used the first three principal components. 

### 4.12. Differential Expression Analysis of miRNAs from Sequencing Reads

To determine the differentially expressed miRNAs, we used the DESeq2 (Bioconductor) library in R code; a library specialized in the analysis of count data that provides methods to evaluate the differential expression using the negative binomial distribution and the shrinkage estimator for the distribution of the variance [116,117]. We normalized the data by calculating each factor’s weight using a function that calculates its weight. With the normalized data, we proceeded to calculate the miRNAs differentially expressed using the “results” function DESeq2 considering differences in expression of those miRNAs with *p*-value adjusted correction multiple, smaller testing 5% [117]. The miRNAs’ differential expression values were visualized in heat maps using the “pheatmap” function in R code [118]; we represented the differential expression data in a grid. Each row represented a miRNA, and each column represented a sample. The squares’ color and intensity represent changes (not absolute values) of the expression. For illustration purposes, unsupervised clustering using the hierarchical clustering based on the Lance-Williams algorithm was performed using the library pheatmap Raivo Kolde (2019). pheatmap: R package version 1.0.12. https://CRAN.R-project.org/package=pheatmap and Cluster Analysis Basics and Extensions. R package version 2.1.0. in R [119,120].

### 4.13. Prediction of Molecular Targets (mRNAs) of microRNAs by TargetScan Mouse

We selected some molecular targets (mRNAs), which had a previous report of changes in their expression in models of decreased growth and differential pigmentation induced by L-Tyr or 5-Brd-2′-dU with B16F1 cells or in other cell models. By using TargetScanMouse 7.1 (http://www.targetscan.org/mmu_71/), we performed the prediction of all potential miRNAs that could have that gene as a molecular target. The report generated was filtered using the list of miRNAs with differential expression obtained by DESeq2 [121].

### 4.14. RT-qPCR-Stem-Loop for miRNAs and RT-PCR for mRNAs

We selected seven miRNAs to confirm their expression by RT-qPCR stem-loop. For this, we used a stable loop-type structure (stem-loop) to provide additional length to the target cDNA, optimize its melting temperature (Tm), and improve the assay’s specificity (Appendix A. List of primer sets for genes and miRNAs used in RT-qPCR and RT-qPCR-stem loop). The miRNAs RT and qPCR assays were conducted with the mirVana™ microRNA Detection Kit (Thermo Fisher Scientific) following the manufacturer’s recommendations. We started with 65 ng of RNA for RT, and cDNA synthesis run in a thermocycler (BIO-RAD, Hercules, California, USA): 30 min at 16 °C, 30 min at 42 °C, and 5 min at 85 °C. For qPCR, we used 2.5 µg of cDNA in the Chromo4™ thermal cycler (BIO-RAD) under the program: 3 min at 95 °C and 40 cycles: 15 s at 95 °C for denaturation, followed by 1 min at 60 °C for ringing and 1 min at 72 °C extensions, after 40 cycles a final extension left at 72 °C for 10 min.

For the RT-qPCR of the mRNAs, we synthesized cDNA from 2 ng of total RNA and 500 µg/mL of OligodT (Invitrogen) and Superscript II reverse transcriptase (Invitrogen); the samples were run in a thermal cycler (BIO-RAD) for 50 min at 42 °C and 15 min at 70 °C. For real-time PCR (qPCR), 600 ng of cDNA, the sense and antisense primers corresponding to each evaluated mRNA (Appendix A. Primers list), and DyNAmo HS SYBR Green Kit (Thermo Fisher Scientific) were used, following the manufacturer’s recommendations. The amplification in a BIO-RAD Chromo4™ under the following conditions: 95 °C 15 min to activate the polymerase and 36 cycles that included 10 s at 96 °C to denature, 30 s annealing temperature according to each set of primers, and 30 s at 72 °C extension. We identified the respective melting curve of 55–95 °C with data obtained from the Opticon Monitor 3 (BIO-RAD) program for both qPCRs of miRNAs and mRNAs. To quantify mRNA and miRNA Differential expression analysis with qPCR, we use the relative expression ratio (rER) model to establish the mRNAs’ miRNAs’ differential expression [122]. This method adjusts the expression level according to the efficiency of each PCR run. As references, we used GAPDH for mRNAs or U6 snRNA for miRNAs.

### 4.15. Construction of Co-Expression Networks for miRNAs (RC-miR)

For the RC-miR construction, we used the previously proposed [123], all in R code. The structure of these networks began with creating a similarity matrix, followed by selecting the similarity threshold, constructing an adjacency matrix, and plotting the generated groupings. It found that the relationship between miRNAs was not linear, so the calculation of the similarity matrix of each of the expression matrices was made with the criterion of mutual information (M.I.) because M.I. is capable of detecting nonlinear correlations that are undetectable for other metrics such as the Pearson coefficient [124]. It was necessary to perform a discretization of the normalized expression data, and we used the “Equal Frequency” method. Finally, the values were adjusted to a correlation coefficient between [0, 1] [125]. We constructed six matrices of similarity by using the M.I. coefficient as a similarity metric and the expression matrix that included the normalized counts of the Small RNA-seq for the miRNAs with differential expression found in the B16F1 cells exposed or not to L-Tyr or 5-Brd-2′-dU, expressed in blocks or by groups: over or under-expressed. We established a similarity threshold to build the adjacency matrix from which the correlation (similarity) between two miRNAs was considered significant. We selected this threshold by comparing the grouping coefficient of the obtained network (*C*(*τ*)) and the expected grouping coefficient for a random network (*Co*(*τv*)), with different values of (tao) *τ* between 0.01 and 0.99 (Equation (1)) [123,126].
(1)$${\;}^{*}=\{\tau_{v}:\left|C\left(\tau_{v}\right)-Cr\left(\tau_{v}\right)|\;>\;\right|C\left(\tau_{v+1}\right)-Cr\left(\tau_{v+1}\right)|\}$$

As the RC-miRs are non-directed networks, we generated an adjacency matrix (A), indicating which nodes were connected and composed only of ones and zeros generated following Equation (2).
(2)$$a_{ii}=0\;for\;all\;ii\;a_{ij}=1\;if\;{|}_{ij}|\;>\;a_{ij}=0\;in\;other\;case$$

We visualized the network using the *igraph* library in R, and we achieved the graphs obtained representing the co-expression networks of miRNAs from multiple experiments.

### 4.16. Regulation Networks and Functional Enrichment

We used miRNet to construct regulatory network models (https://www.mirnet.ca/), an open-access web-based tool that integrates multiple statistical tools, data mining, and visualization systems miRNA-target interaction studies [127,128]. In addition to implementing a flexible interface to filter, refine, and personalize data during network creation, miRNet contains a network visualization system with the possibility of functional enrichment analysis via KEGG, G.O., and Reactome signaling pathways. Tool fed with miRNAs I.D.s over or under-expressed by exposure to L-Tyr or 5-Brd-2′-dU, and for functional enrichment, we used the KEGG signaling pathways and statistical analysis hypergeometric test.

### 4.17. Statistic Analysis

We used an unpaired Student’s *t*-test to test for significant differences between each exposure and control. Except where otherwise noted, the number of independent replicates used in the statistical analyses was three; values reported as mean ± S.D. and results were considered not significant (ns) with *p* > 0.05, significant (*) with *p* < 0.05, very significant (**) with *p* < 0.01, highly significant (***) with *p* < 0.001 and very highly significant (****) with *p* < 0.0001. For multiple tests, the significance level (a two-tailed *t*-multiple test was performed *), and the differences were considered statistically significant for a *p*-value < 0.05 using the Holm–Sidak method with GraphPad Prism^®^ (Graphpad Software Inc., La Jolla, CA, USA). The functional enrichment by KEGG signaling pathways in the miRNet web tool included hypergeometric tests and was considered statistically significant for *p* < 0.05 values. Finally, the significance statistic of differentially expressed microRNAs from small RNAseq counts was carried out with DESeq2 algorithm in R code and included the correction of multiple tests.

## 5. Conclusions

L-Tyr and 5-Brd-2′-dU induces a profile of microRNAs in the melanoma cell line B16F1. We identified potential miRNA antagonism and coordination networks on genes involved in pigmentation, cell cycle control, and senescence.

## Figures and Tables

**Figure 1 ijms-22-01591-f001:**
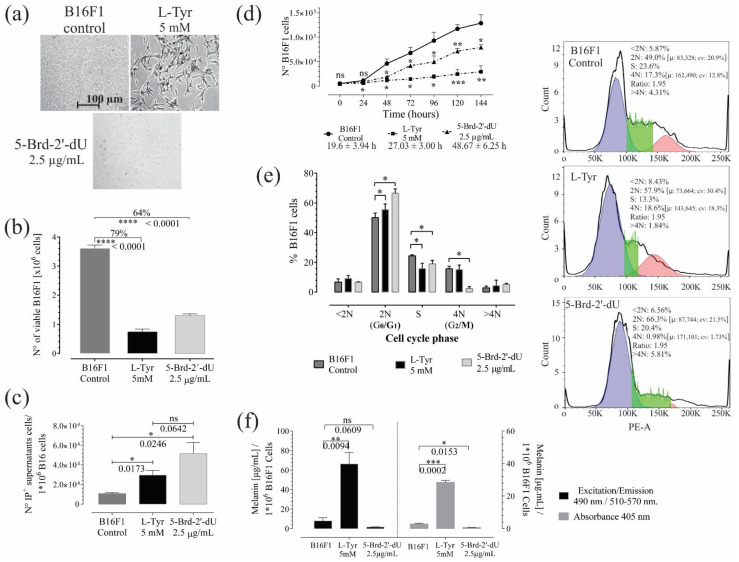
Exposure to L-Tyr or 5-Brd-2′-dU for 72 h in B16F1 cells generates a reduction in the number of cells and affects melanin concentration. (**a**) Representative photographs of B16F1 cells exposed to 5 mM L-Tyr or 2.5 μg/mL 5-Brd-2′-dU after 72 h. (**b**) Quantification of the number of viable cells by Trypan Blue exclusion assay. (**c**) The number of cells in supernatants that incorporate Propidium Iodide (I.P.). (**d**) Changes in B16F1 cell number by MTT assay and population doubling times. (**e**) Frequency histograms of DNA content. Permeable cells incorporated PI; the cell cycle analysis corresponds to a univariate Gaussian distribution model. FlowJo algorithm function revealed phase S cells, (**f**) Melanin concentration from B16F1 cells after exposure to L-Tyr or 5-Brd-2′-dU by spectrophotometry fluorescence. The significance (*) using two-tailed multiple *t*-tests, and the differences were considered statistically significant for a *p*-value < 0.05 using Welch’s correction. significant (*) with *p* < 0.05, very significant (**) with *p* < 0.01, highly significant (***) with *p* < 0.001 and very highly significant (****) with *p* < 0.0001.

**Figure 2 ijms-22-01591-f002:**
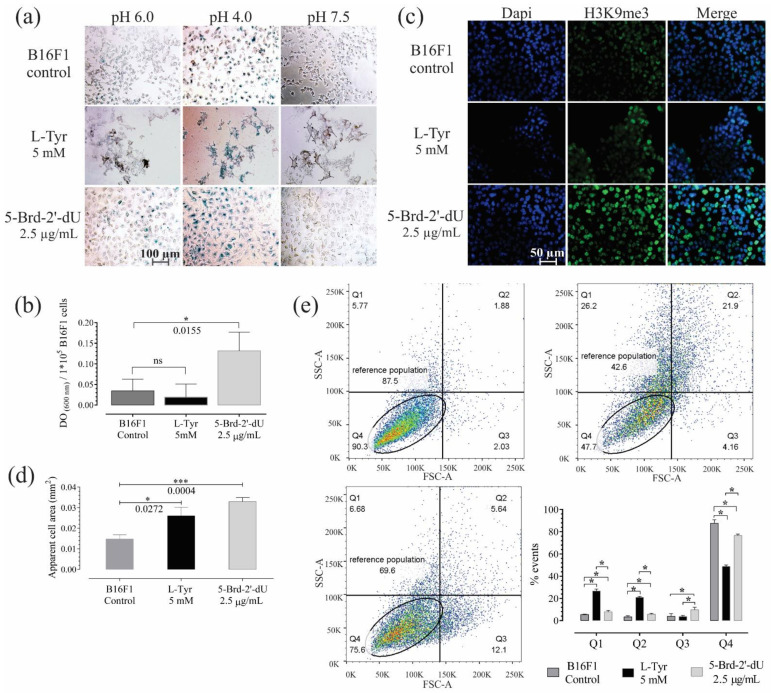
B16F1 cells exposed for 72 h to 5-Brd-2′-dU reveal changes in markers associated with senescence. (**a**) Representative photographs of B16F1 cells after the chromogenic assay for senescence-associated β-Galactosidase activity (S.A. β-Gal). (**b**) Quantification by spectrophotometry of the S.A. β-Gal activity at pH 6.0. (**c**) Representative photographs of immunofluorescence (IF) in B16F1 cells labeled against histone H3K9me3 (green) and nucleus staining with DAPI (blue). (**d**) Measurement of the apparent cell area in µm2 from bright-field photographs of 100 B16F1 cells (*n* = 3). (**e**) Diagram of cell complexity (SSC-A) and size (FSC-A) acquired by flow cytometry and its distribution by quartiles (Q1, Q2, Q3, and Q4). The significance level (*) using two-tailed multiple *t*-tests, and the differences were considered statistically significant for a *p*-value < 0.05 using the Holm–Sidak method. Significant (*) with *p* < 0.05, significant (***) with *p* < 0.001.

**Figure 3 ijms-22-01591-f003:**
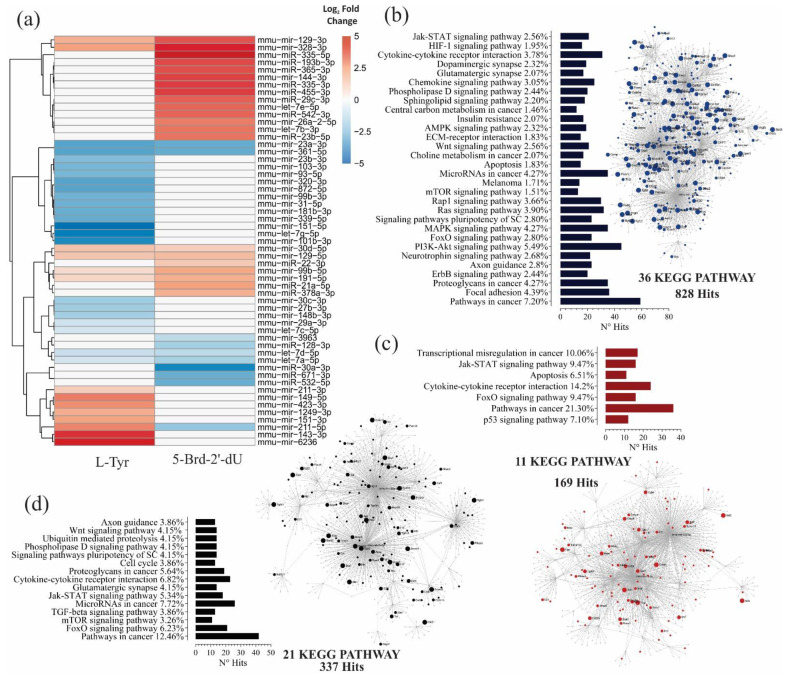
B16F1 cells induced to differential pigmentation and decreased proliferation, show changes in miRNA expression and, together with their molecular targets, make up potential gene interaction networks. (**a**) A heat map with the relative expression levels of 55 differentially expressed miRNAs, obtained by Small RNA-seq from B16F1 cells exposed to 5 mM L-Tyr or 2.5 μg.mL^−1^ 5-Brd-2′-dU concerning non-exposed B16F1 control cells; red, expression up, blue, expression down, and white indicator of no variation. Differential expression by using the DESeq2 program. (**b**–**d**) Analysis of functional enrichment by KEGG signaling pathways of the potential interactions between differentially expressed miRNAs and their molecular targets in a network model, for exposure of melanoma cells to L-Tyr (**b**), 5-Brd-2′-dU (**c**), and 5-Brd-2′-dU with relative to L-Tyr (**d**). The related KEGG signaling pathways were considered statistically significant for a *p*-value < 0.05 after applying a hypergeometric test.

**Figure 4 ijms-22-01591-f004:**
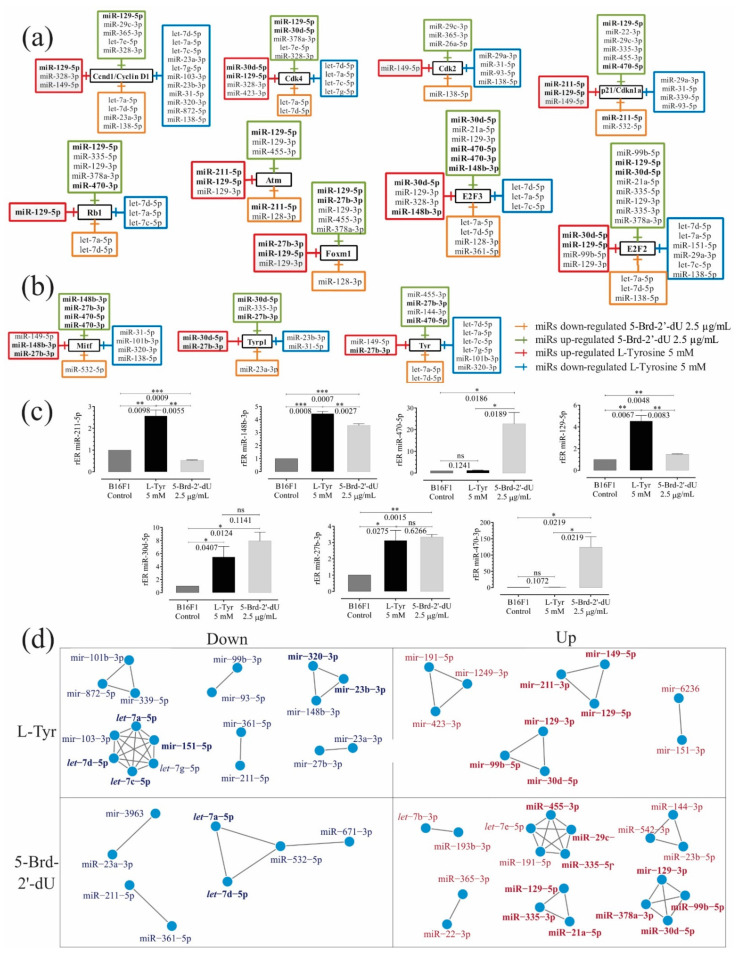
Sets of miRNAs show coordinated expression patterns in the differential pigmentation model and reduced proliferation induced in melanoma B16F1 cells. Sets of over-and under-expressed miRNAs and their potential candidate genes associated with cell cycle and senescence control (**a**) and melanogenesis (**b**) using the TargetScanMouse tool version 7.1. (**c**) Relative expression radius (rER) of miRNAs by RT-qPCR stem-loop for miRNAs 211-5p, 148b-3p, 470-5p, 129-5p, 27b-3p, 470-3p and 30d-5p (*n* = 3). (**d**) Co-expression networks from normalized Small RNA-Seq count of under-expressed miRNAs of B16F1 cells exposed to L-Tyr or 5-Brd-2′-dU. The level of significance (*) using two-tailed multiple *t*-tests. The differences were considered statistically significant for a *p*-value < 0.05, ns: not significant. Significant (*) with *p* < 0.05, very significant (**) with *p* < 0.01, highly significant (***) with *p* < 0.001.

**Figure 5 ijms-22-01591-f005:**
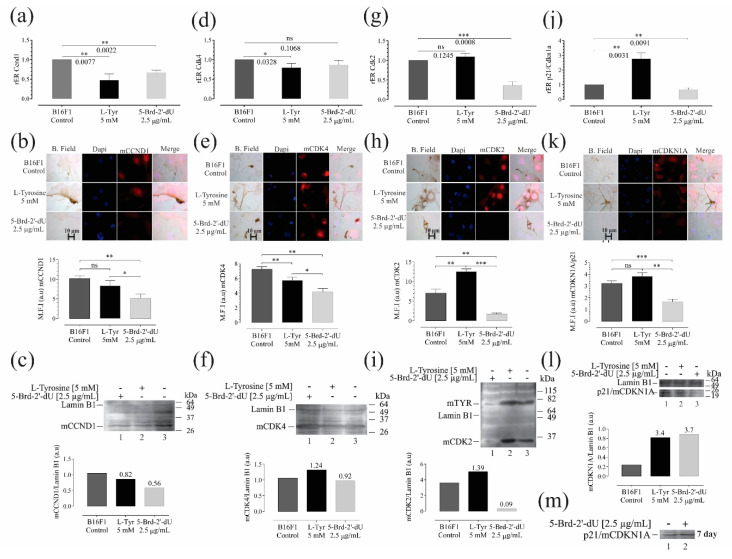
Variations in the expression of genes and their protein products, associated with controlling the cell cycle and senescence in B16F1 cells, induced differential pigmentation and decreased proliferation. Relative expression radius (rER) by RT-qPCR, (*n* = 4), representative photographs of the location and distribution by immunofluorescence (red) in nuclear counterstain with DAPI (blue), quantification of expression by mean fluorescence intensity (MFI), and western blot for CCnd1 (**a**–**c**), Cdk4 (**d**–**f**), Cdk2 (**g**–**i**) and p21 (**j**–**m**). The level of significance (*) using two-tailed multiple *t*-tests. The differences were considered statistically significant for a *p*-value < 0.05, ns: not significant. Significant (*) with *p* < 0.05, very significant (**) with *p* < 0.01, highly significant (***) with *p* < 0.001.

**Figure 6 ijms-22-01591-f006:**
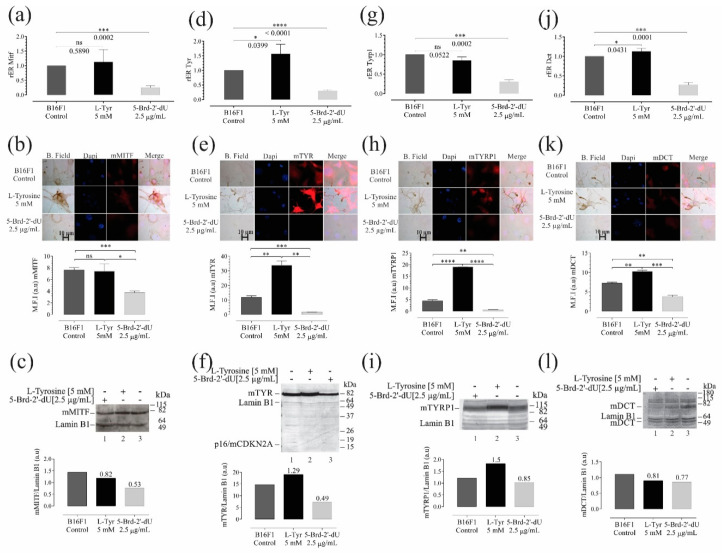
Variations in the expression of genes and their protein products are associated with melanogenesis in B16F1 cells induced at differential pigmentation and a decrease in proliferation. Relative expression radius (rER) by RT-qPCR, *n* = 4, representative photographs of the location and distribution by immunofluorescence (red) in nuclear counterstain with DAPI (blue), quantification of expression by mean fluorescence intensity (MFI), and western blot for the master regulator of melanocyte cell differentiation—MITF (**a**–**c**), and the enzymes Tyr (**d**–**f**), Trp1 (**g**–**i**), and Dct (**j**–**l**). The level of significance (*) using two-tailed multiple *t*-tests. The differences were considered statistically significant for a *p*-value < 0.05, ns: not significant. Significant (*) with *p* < 0.05, very significant (**) with *p* < 0.01, highly significant (***) with *p* < 0.001 and very highly significant (****) with *p* < 0.0001.

**Table 1 ijms-22-01591-t001:** List of differential microRNAs expression in B16F1 melanoma cells exposed for 72 h to 5-Bromo-2-Deoxyuridine 2.5 µg.mL^−1^ or L-Tyrosine amino acid 5 mM. microRNA ID refers to the identification number of miRNA register in miRBase; Log2 Fold Change represents times of change in the expression; *p*-value corresponds to the statistical significance and p-*adj* significance by multiple tests correction; baseMean means the average counts.

L-Tyr vs. Respect to Unexposed B16 Cells					5-Brd-2′-dU vs. Respect to Unexposed B16 cells					5-Brd-2′-dU vs. Respect to L-Tyr				
microRNA ID	log_2_ Fold Change	*p* _Value_	p*_adj_*	BaseMean	microRNA ID	log_2_ Fold Change	*p* _Value_	p*_adj_*	BaseMean	microRNA ID	log_2_ Fold Change	*p* _Value_	p*_adj_*	BaseMean
mmu-let-7d-5p	−0.93	8.701 × 10^−4^	2.504 × 10^−2^	2014	mmu-mir-3963	−1.85	1804 × 10^−9^	1.401 × 10^−6^	29645	mmu-mir-30d-5p	−1.04	8.70 × 10^−6^	6.50 × 10^−4^	2326
mmu-let-7a-5p	−0.64	1.05 × 10^−3^	3.27 × 10^−2^	1648	mmu-let-7a-5p	−0.96	6.47 × 10^−4^	2.093 × 10^−2^	1300	mmu-mir-211-5p	−2.20	1.03 × 10^−5^	7.15×10^−4^	2024
mmu-mir-151-5p	−0.94	3.567 × 10^−4^	1.375 × 10^−2^	859	mmu-let-7d-5p	−0.89	1.425 × 10^−3^	3.882 × 10^−2^	1030	mmu-mir-1249-3p	−2.23	6.09 × 10^−5^	3.02 × 10^−3^	705
mmu-mir-29a-3p	−0.94	2.239 × 10^−7^	3.885 × 10^−5^	636	mmu-mir-211-5p	−1.36	3.164 × 10^−4^	1.199 × 10^−2^	696	mmu-mir-211-3p	−2.87	1.16 × 10^−6^	1.89 × 10^−4^	126
mmu-let-7c-5p	−0.84	1.247 × 10^−3^	3.762 × 10^−2^	533	mmu-mir-128-3p	−1.91	9.496 × 10^−5^	4.589 × 10^−3^	232	mmu-mir-30a-3p	−5.10	5.56 × 10^−9^	2.70 × 10^−6^	45
mmu-mir-27b-3p	−1.47	5.270 × 10^−4^	1.97 × 10^−2^	377	mmu-mir-361-5p	−1.72	8.53 × 10^−5^	4.014 × 10^−3^	204	mmu-mir-501-3p	−3.40	1.38 × 10^−4^	6.27 × 10^−3^	43
mmu-mir-148b-3p	−2.00	2.046 × 10^−5^	1.775 × 10^−3^	118	mmu-mir-138-5p	−3.26	1.889 × 10^−5^	2.000 × 10^−4^	65	mmu-mir-130b-3p	−3.16	5.74 × 10^−4^	2.15 × 10^−2^	31
mmu-mir-23a-3p	−3.05	1.413 × 10^−4^	7.519×10^−3^	49	mmu-mir-671-3p	−2.35	1.686 × 10^−3^	4.292 × 10^−2^	27	mmu-mir-671-3p	−2.76	1.30 × 10^−4^	6.12 × 10^−3^	29
mmu-mir-99b-3p	−2.98	3.64 × 10^−7^	5.535 × 10^−5^	45	mmu-mir-30a-3p	−4.20	1.719 × 10^−6^	2.670 × 10^−4^	26	mmu-mir-351-5p	−4.35	9.37 × 10^−6^	9.10 × 10^−4^	25
mmu-let-7g-5p	−4.44	8.90 × 10^−9^	4.118 × 10^−6^	45	mmu-mir-23a-3p	−3.13	8.000 × 10^−4^	2.484 × 10^−2^	17	mmu-mir-6236	−4.22	1.52 × 10^−6^	2.02 × 10^−4^	22
mmu-mir-30c-3p	−2.09	6.679 × 10^−4^	2.377 × 10^−2^	35	mmu-mir-532-5p	−3.10	1.498 × 10^−3^	3.976 × 10^−2^	11	mmu-mir-103-3p	−3.20	1.32 × 10^−3^	4.38 × 10^−2^	11
mmu-mir-103-3p	−2.87	5.338 × 10^−5^	4.116 × 10^−3^	32	mmu-mir-99b-5p	1.14	2.312 × 10^−4^	7.241 × 10^−3^	10602	mmu-mir-26a-5p	0.73	6.91 × 10^−5^	3.63 × 10^−3^	1879
mmu-mir-23b-3p	−2.70	1.346 × 10^−4^	7.473 × 10^−3^	28	mmu-mir-129-5p	1.6	2.962 × 10^−5^	1.550 × 10^−3^	1955	mmu-mir-21a-5p	1.67	6.68 × 10^−5^	3.02 × 10^−3^	518
mmu-mir-361-5p	−3.39	1.787 × 10^−4^	8.670 × 10^−3^	23	mmu-mir-22-3p	1.32	1.698 × 10^−4^	6.194 × 10^−3^	764	mmu-let-7b-3p	1.78	9.39 × 10^−5^	4.57 × 10^−3^	203
mmu-mir-31-5p	−3.01	1.881 × 10^−4^	8.704 × 10^−3^	22	mmu-mir-30d-5p	1.20	2.214 × 10^−5^	1.719 × 10^−3^	618	mmu-mir-335-5p	4.00	3.35 × 10^−4^	1.14 × 10^−2^	141
mmu-mir-101b-3p	−4.12	3.302 × 10^−6^	3.554 × 10^−4^	17	mmu-mir-191-5p	1.48	3.663 × 10^−4^	1.326 × 10^−3^	574	mmu-mir-193b-3p	6.14	1.23 × 10^−14^	8.95 × 10^−12^	68
mmu-mir-320-3p	−3.37	1.658 × 10^−4^	8.670 × 10^−3^	14	mmu-mir-21a-5p	2.17	2.679 × 10^−4^	7.450 × 10^−3^	560	mmu-mir-484	2.88	1.89 × 10^−5^	1.44 × 10^−3^	65
mmu-mir-181b-3p	−3.11	7.574 × 10^−4^	2.564 × 10^−2^	12	mmu-mir-335-5p	3.32	1.029 × 10^−4^	3.727 × 10^−3^	329	mmu-mir-99b-3p	2.89	3.87 × 10^−4^	1.52 × 10^−2^	54
mmu-mir-872-5p	−3.37	2.226 × 10^−4^	9.848 × 10^−3^	12	mmu-let-7b-3p	1.77	1.009 × 10^−3^	2.909 × 10^−2^	228	mmu-mir-335-3p	4.75	1.05 × 10^−5^	6.80 × 10^−4^	51
mmu-mir-29a-3p	−3.51	1.083 × 10^−4^	6.592 × 10^−3^	11	mmu-mir-129-3p	2.10	2.947 × 10^−4^	1.149 × 10^−2^	191	mmu-mir-365-3p	4.30	9.47 × 10^−3^	7.20 × 10^−4^	42
mmu-mir-339-5p	−3.15	6.087 × 10^−4^	2.283 × 10^−2^	11	mmu-mir-335-3p	3.46	4.762 × 10^−5^	2.917 × 10^−3^	171	mmu-miR-144-3p	3.54	4.52 × 10^−4^	1.74 × 10^−2^	35
mmu-mir-93-5p	−2.84	1.31 × 10^−3^	3.868 × 10^−2^	10	mmu-mir-455-3p	2.75	2.150 × 10^−4^	7.299 × 10^−3^	153	mmu-mir-27b-3p	3.46	6.08 × 10^−5^	3.17 × 10^−3^	32
mmu-mir-30d-5p	1.02	8.078 × 10^−6^	7.626 × 10^−4^	5118	mmu-mir-378a-3p	2.07	5.965 × 10^−5^	3.431 × 10^−3^	104	mmu-mir-320-3p	3.65	2.18 × 10^−4^	9.65 × 10^−3^	23
mmu-mir-211-5p	1.1	6.743 × 10^−4^	2.318 × 10^−2^	1391	mmu-mir-193b-3p	3.64	1.059 × 10^−4^	4.269 × 10^−3^	97	mmu-mir-27a	4.21	1.61 × 10^−5^	1.38 × 10^−3^	21
mmu-mir-211-3p	1.26	8.183 × 10^−5^	5.408 × 10^−3^	159	mmu-miR-144-3p	3.79	9.074 × 10^−5^	4.145 × 10^−3^	42	mmu-mir-193b	4.03	4.33 × 10^−5^	2.52 × 10^−3^	20
mmu-mir-129-5p	1.35	4.746 × 10^−4^	1.553 × 10^−2^	677	mmu-mir-29c-3p	3.42	2.991 × 10^−4^	1.161 × 10^−2^	23	mmu-mir-221-3p	3.39	7.87 × 10^−4^	2.80 × 10^−2^	15
mmu-mir-99b-5p	0.85	2.270 × 10^−4^	9.848 × 10^−3^	670	mmu-mir-365-3p	3.64	7.318 × 10^−5^	3.788 × 10^−3^	22	mmu-let-7e-5p	3.66	2.37 × 10^−4^	1.02 × 10^−2^	13
mmu-mir-1249-3p	2.11	6.548 × 10^−8^	1.515 × 10^−5^	524	mmu-mir-26a-2-5p	3.03	1.511 × 10^−3^	3.976 × 10^−2^	22	mmu-mir-455-3p	3.36	8.42 × 10^−4^	2.90 × 10^−2^	12
mmu-mir-191-5p	1.00	3.067 × 10^−4^	1.228 × 10^−2^	355	mmu-let-7e-5p	3.35	4.058 × 10^−4^	1.500 × 10^−2^	20					
mmu-mir-129-3p	2.10	3.328 × 10^−6^	3.554 × 10^−4^	322	mmu-mir-328-3p	4.18	8.692 × 10^−6^	1.038 × 10^−3^	16					
mmu-mir-328-3p	2.50	6.864 × 10^−5^	5.014 × 10^−3^	81	mmu-mir-542-3p	3.37	5.392 × 10^−4^	1.820 × 10^−2^	16					
mmu-mir-151-3p	2.38	1.237 × 10^−4^	7.153 × 10^−3^	61	mmu-mir-23b-5p	3.13	1.360 × 10^−3^	3.770 × 10^−2^	15					
mmu-mir-143-3p	4.01	6.562 × 10^−7^	8.280 × 10^−5^	40										
mmu-mir-149-5p	3.03	9.253 × 10^−4^	2.987 × 10^−2^	26										
mmu-mir-6236	4.38	1.794 × 10^−7^	3.556 × 10^−5^	21										
mmu-mir-423-3p	2.79	1.246 × 10^−^^3^	3.762 × 10^−2^	20										

## Data Availability

Publicly available datasets were analyzed in this study. This data can be found here [https://www.ncbi.nlm.nih.gov/geo/query/acc.cgi?acc=GSE147170].

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
