# Peer review of "Reciprocal Changes in miRNA Expression with Pigmentation and Decreased Proliferation Induced in Mouse B16F1 Melanoma Cells by l-Tyrosine and 5-Bromo-2′-Deoxyuridine"

_ijms, 2021, doi:10.3390/ijms22041591_

Round 1
Reviewer 1 Report
Rivera et al highlighted that specific miRNA signatures (including 470-3p, 470-5p, 30d-5p, 129-5p, 148b-3p, 27b-3-3p, and 211-5p) and their potential targets (including MITF, Tyr, Tyrp1, Cyclin D1, Cdk2, Cdk4, p21, and p27) were associated with the progression of melanoma. The authors used several molecular biology, biochemistry and protein experiments and bioinformatics analyses to assess and evaluate the miRNA changes in melanoma cells exposing to L-Tyr and 5-Brd-2’-dU. The overall findings facilitate the deeper understanding into the mechanisms of tumour progression.
Specific comments:
- Figure 3a, hierarchical cluster 3 and state the method of clustering in the methods section. Colour legend lacks of labels. Is that log2 gene expression? For all the heatmaps in the manuscript, the method used for clustering should be included in the manuscript. Additionally, the clustering tree (dendrogram) should be included in the heatmaps. Could the authors advise?
- Figure 3b,c,d, may be good for conference/seminar presentations. However, they served no purpose for the manuscript. Could the authors clarify the purpose of the signalling network hubs that were presented in the figure? The different sizes of the dots (nodes) in the network may represent the expression, however, the sizes of the nodes were not described. In addition, the purpose of the network graphs were used to highlight the interaction between miRNA and mRNA. However, the fonts were too small to be read, and unable to interpret the interaction. Could the authors clarify? If authors insisted in putting the network graphs, then an interaction 3D network should be included in the manuscript with a HTML link so that researchers in the same field can explore further.
- Figure 4d may require better visualisation. Some of the fonts are masked by the lines (edges). Could the authors clarify the meaning of the different length of the edges?
- The correlation plot/analysis between miRNAs of interests and their mRNAs, and miRNAs and proteins should be included in the manuscript. This will further highlight the downstream effect for miRNAs.
- For Figure 6i, could the authors provide the editor/reviewers the full gel image, it appears that the gel image was cropped and pasted due to the colour contrast differences. Could the authors clarify?
- In the Discussion, the authors mentioned that the technical effects of NGS, microarray and RT-PCR may have affected the differential expression analysis. However, the likely cause may be due to the limited number of samples (lack of power for analysis) and the coverage of sequencing is not sufficient for their expected dysregulated miRNAs/genes. Could the authors clarify?
Minor comments:
- Typographical and grammar errors. Could the authors provide the corrected version for publication? For example, line 396, 436, 473 etc.
- Suggest authors to put Table 1 primers and other relevant detailed methods as supplementary methods.
- Table 2 should be as a supplementary table. Replace “comma” with “dot”. For example, 8.701E-04
Reviewer 2 Report
The manuscript entitled "Reciprocal changes in miRNA expression with pigmentation and decreased proliferation induced in mouse B16F1 melanoma cells by L-Tyrosine and 5-Bromo-2'-deoxyuridine" addresses miRNA expression alterations in L-Tyrosine and 5BrDU treated B16F1 melanoma cells. The authors perform RNAseq to identify differentially expressed miRNAs and check their involvement by rt-PCR and follow the expression of potential targets by mRNA and protein expression analysis. Together, these analyses allow for insight into miRNA expression changes that might mediate the anti-proliferative, pro-senescence effects of the L-Tyr and 5BrDU.
The work is well performed, technically sound and exhaustive. The paper is well written. The initial part of the introduction is excessively general and descriptive. The discussion section could benefit from considerable shortening.
However, there is a very important flaw in the work: the use of a single cell line does not allow for any generalization of the effects. Essentially, we are interested in blocking melanoma growth, not B16F1 growth. B16F1 cells are a widely used model that shows, however, many particularities as for example a very extreme metabolism that limits its model character. It is therefore absolutely necessary to confirm the findings in other melanoma cells, especially in primary human melanoma cultures.
Reviewer 3 Report
The authors try to discuss in the manuscript if reciprocal changes in miRNA expression with pigmentation, and decreased proliferation are induced in B16F1 melanoma after L-tyrosine and 5-Brd-2’-dU treatment. The manuscript is well prepared and very interesting, but minor revision must be performed:
INTRODUCTION
Line 66 – something is wrong with 5-Brd-2’-dU unit, please correct it, probably it should be 2.5 µg/mL.
RESULTS
Fig.1 - Fig 1 b, c, and especially e must be corrected since the size font is too small. I suggest changing the name of the axis of Fig 1b by adding units in square bracket [x106 cells] and then on the axis left only the numbers from 1 to 4. 3 graphs from Figure 1 e are unreadable. Moreover, in the materials and methods section must be add point entitled statistical analysis, with the description of what method/methods were used, the number of repeats, what was calculated SD or SEM, and what software was used. From figure 1 description is known that the Authors used a t-test, which is not appropriate since one-way Anova and/or two-way Anova have to be used. Please prepare a new analysis and make corrections in the manuscript and on Figures since not all results will be still significant.
Fig 2 – the same problem with Fig 2e – 3 graphs are unreadable. Moreover, in Fig 2d the axis described as the apparent cell area in µm2 why the values are present as the number of cells, not as area? If the authors marked 1x104 µm2 it is equal to 0.01 mm2, why not use mm2 instead of µm2 if the values are so high?
Table 2 – font must be changed for bigger since the table is not readable.
Fig 3 – Please change the font of Figs b,c, and d for the same size as Fig 1a.
Fig 4 – Font of Fig 4 a and b must be the same size as Fig 4c
MATERIALS & METHODS
Point 4.8 – some information is missing: what was the concentration of use primary and secondary antibodies?
REFERENCES
The references must be corrected according to the Journal format (some parts must be bold, page range must be full e.g.103-121 not 103-21, etc.).
Round 2
Reviewer 2 Report
The authors have only partially replied to my comments. I still believe that experimental studies on a single cell line do not allow for any generalization given the high biological variability. This particularly applies to the B16 models since these cell lines cannot be taken as a general model for melanoma. The fact that the same authors have reported on other cell lines in other publications, a part from raising the concern that the present paper might be a siple replication study, does not help since the aspect of miRNA involvement was not addressed.
The main points should therefore be addressed through the analysis of at oeast one different cell line.
Author Response
Response to Reviewer 2 Comment
We thank reviewer 2 for the comment:
"The authors have only partially replied to my comments. I still believe that experimental studies on a single cell line do not allow for any generalization given the high biological variability. This particularly applies to the B16 models since these cell lines cannot be taken as a general model for melanoma. The fact that the same authors have reported on other cell lines in other publications, a part from raising the concern that the present paper might be a siple replication study, does not help since the aspect of miRNA involvement was not addressed.
The main points should therefore be addressed through the analysis of at oeast one different cell line".
Response to comment: We agree with the reviewer that experimental studies on a single cell line do not allow any generalization given the high biological variability. The results apply only to the murine melanoma cell line B16 under specific experimental conditions.
We considered it necessary to report these results because determining those miRNAs expressed in the growth decreased phenotype in mouse melanoma cell line B16; we compared the same cell line under two different phenotypes (biologic behavior) from the same cell population with the same genetic background. While a basis for comparison is still maintained, many differences between cell lines and primary cell cultures and the heterogeneity in melanoma from biopsies or surgical specimens may limit these findings' relevance to the miRNA expression. They may be different and questionable for the genetic variations.
We cannot extrapolate the results found in our work to another melanoma model and confirm the findings in other melanoma cells is warranted, especially in primary human melanoma cultures. We included the reviewer's comment on the limitation of the model used in the manuscript's discussion section (lanes 512-513).